ated">

# Functional annotation of the animal genomes: An integrated annotation resource for the horse

Sichong Peng[1], Anna R. Dahlgren[1], Callum G. Donnelly[1], Erin N. Hales[1], Jessica L. Petersen[2], Rebecca R. Bellone[1,3], Ted Kalbfleisch[4], Carrie J. Finno[1]*

1 Department of Population Health and Reproduction, School of Veterinary Medicine, University of California-Davis, Davis, California, United States of America, 2 Department of Animal Science, University of Nebraska—Lincoln, Lincoln, Nebraska, United States of America, 3 Veterinary Genetics Laboratory, School of Veterinary Medicine, University of California-Davis, Davis, California, United States of America, 4 Gluck Equine Research Center, Department of Veterinary Science, University of Kentucky, Lexington, United States of America

* cjfinno@ucdavis.edu

**Data Availability Statement:** RNA-seq data can be accessed from ENA and SRA under the accession number PRJEB26787 (female tissues) and PRJEB53382 (male tissues). Iso-seq data can be

## Abstract

The genomic sequence of the horse has been available since 2009, providing critical resources for discovering important genomic variants regarding both animal health and population structures. However, to fully understand the functional implications of these variants, detailed annotation of the horse genome is required. Due to the limited availability of functional data for the equine genome, as well as the technical limitations of short-read RNA-seq, existing annotation of the equine genome contains limited information about important aspects of gene regulation, such as alternate isoforms and regulatory elements, which are either not transcribed or transcribed at a very low level. To solve above problems, the Functional Annotation of the Animal Genomes (FAANG) project proposed a systemic approach to tissue collection, phenotyping, and data generation, adopting the blueprint laid out by the Encyclopedia of DNA Elements (ENCODE) project. Here we detail the first comprehensive overview of gene expression and regulation in the horse, presenting 39,625 novel transcripts, 84,613 candidate cis-regulatory elements (CRE) and their target genes, 332,115 open chromatin regions genome wide across a diverse set of tissues. We showed substantial concordance between chromatin accessibility, chromatin states in different genic features and gene expression. This comprehensive and expanded set of genomics resources will provide the equine research community ample opportunities for studies of complex traits in the horse.

## Author summary

Functional annotation of a reference genome provides critical information that pertains the tissue-specific gene expression and regulation. Non-model organisms often rely on existing annotations of human and mouse genomes and the conservation between species for their genome annotation. This approach has limited power in annotating transcripts

accessed from ENA and SRA under the accession number PRJEB53020. ATAC-seq data can be accessed from ENA and SRA under the accession number PRJEB53037 (https://data.faang.org/dataset/PRJEB53037). Histone modification peaks can be accessed from FAANG data portal (https://data.faang.org/) under the accession number PRJEB35307 (https://data.faang.org/dataset/PRJEB35307). CTCF ChIP-seq data can be accessed from SRA/ENA under project accession PRJEB41079. Histone ChIP-seq data were published by Kingsley et al. (https://doi.org/10.3390/genes11010003) and Barber et al (thesis; https://digitalcommons.unl.edu/animalscidiss/233/). The GO ontology database link used for the pathway analyses can be found at 10.5281/zenodo.6399963 and was released on March 3, 2022. The integrated track hub hosted on UCSC genome browser can be accessed at https://genome.ucsc.edu/s/cjfinno/equCab3. All data underlying this track hub can be accessed via UCSC table browser or at https://github.com/FinnoLab/FAANGtracks/tree/main/data.

**Funding:** This project was supported by the Grayson-Jockey Club Research Foundation (C.J.F., J.L.P, R.R.B.), USDA National Institute of Food and Agriculture Animal Breeding and Functional Annotation of Genomes (A1201) Grant 2019-67015-29340 (J.L.P, C.J.F., R.B.) as well as NRSP-8 Species Coordinator Funds from the USDA National Institute of Food and Agriculture (C.J.F., J.L.P, R.R.B.), and the UC Davis Center for Equine Health with funds provided by the State of California pari-mutuel fund and contributions by private donors (C.J.F., J.L.P, R.R.B.). Additional support for C.J.F. was provided by NIH NCATS L40 TR001136. The funders had no role in study design, data collection and analysis, decision to publish, or preparation of the manuscript.

**Competing interests:** The authors have declared that no competing interests exist.

and regulatory elements that are less evolutionarily conserved. Such are the cases of alternatively spliced isoforms and enhancer elements. In a large-scale collaborated effort, Functional Annotation of Animal Genome (FAANG) aims to generate species-specific and tissue-aware functional annotation for farm animals. In this study, we present the overall annotation efforts and findings from the equine FAANG group. This integrated annotation for the horse genome provides, for the first time, a comprehensive overview of gene expression and regulation landscape in nine major equine tissues, as well as an analytical framework for further inclusion of other important tissues.

## Introduction

A reference genome for the horse has been available since 2009 [1], with an improved assembly EquCab3.0 available since 2018 [2]. EquCab3.0 contains 3,771 gaps comprising 9 Mb (0.34% of the genome) and has a scaffold N50 of 86 Mb, with 99.7% mammalian Benchmark Universal Single-Copy Orthologs (BUSCO). This high-quality assembly has enabled development of critical tools and many important discoveries in the horse, which were reviewed by Raudsepp *et al.* [3].

Accompanying the reference genome, annotation was made available via the RefSeq [4] and Ensembl [5] annotation pipelines. The latest RefSeq annotation for EquCab3.0 contains 33,146 genes, of which 21,129 are protein coding, with an average isoform-to-gene ratio of 2.3 [6]. The Ensembl annotation contains 29,969 genes, of which 20,955 are protein coding, with an average isoform-to-gene ratio of 2.0 [7]. With limited public mRNA-seq data for the horse, both RefSeq and Ensembl annotation relied heavily on computational prediction and comparative genomics by translating human and mouse annotation to the horse genome. While this approach produced high-quality annotation for most highly conserved protein-coding genes, it was not able to accurately identify many alternate splicing (AS) in multi-exonic genes. This is evident when comparing the isoform-gene ratios annotated in the horse genome (2.3 and 2.0 in RefSeq and Ensembl, respectively) to that annotated in the human genome (4.0) [8]. While this difference can be attributed to vast quantities of transcriptomic data available in human, recent developments in the long-read sequencing technology provided a unique opportunity for non-model organisms to quickly annotate AS without generating a prohibitively large amount of data. Alternative splicing has been shown to drive cell differentiation and tissue-specific functions [9] and variants leading to aberrant AS have been associated with many diseases [10]. While *in-silico* tools exist to predict variant-induced alterations in AS, accurate annotation of AS isoforms is necessary to establish a reference [11]. Recent advances in long-read sequencing technologies have enabled new approaches to experimentally categorize AS across tissues [12]. In particular, Iso-seq has been successfully applied to various species to characterize AS isoforms [13–15].

In eukaryote genomes, DNA is organized in a three-dimensional structure, where nucleosomes are dynamically unpacked in actively transcribed or regulatory regions [16–19]. This dynamic chromatin remodeling constitutes a crucial aspect of gene regulation: cis-regulatory elements are brought near their target regions by formation of chromatin loops and transcription factors (TF) are recruited to exposed DNA elements. Genetic variants altering this regulatory landscape have been demonstrated to have phenotypic effects [20–22]. Therefore, annotating the genome by specifically defining these cis-regulatory elements (CREs) can provide significant context to understanding genetic variations contributing to many important traits in the horse.

Active CREs are typically characterized by a lack of nucleosome binding and therefore, chromatin accessibilities are often used as a proxy for identifying active regulatory elements [23]. The assay for transposase-accessible chromatin using sequencing (ATAC-seq) is a popular method to assess the genome-wide chromatin accessibilities, owing to its simple protocol and quick turn-around time [24]. Several efforts have been made to adapt the original ATAC-seq protocol to tissue [25] and cryopreserved nuclei [26] samples. We previously demonstrated the feasibility of interrogating genome-wide chromatin accessibility using both flash frozen tissues as well as cryopreserved nuclei in the horse [27].

While the complex molecular mechanism through which this process is regulated remains an active field of research, a growing body of evidence points to histone protein post-translational modifications as an important intermediary of transcription regulation [28–30]. Specifically, histone protein 3 lysine 4 mono- and tri-methylation (H3K4me1 and H3K4me3) have been shown to be enriched around the enhancer and promoter regions, respectively [31,32], with known downstream effectors that further regulate gene expression [33–35]. Additionally, H3K27ac is enriched around active elements and associated with higher levels of gene expression [36]. On the other hand, H3K27me3 is usually found around genes that are not active [37]. Taken together, these protein modifications can be strong indicators of functional activities in specific genomic regions.

To improve AS annotation for the horse transcriptome and to systemically categorize these epigenetic features and identify potential CREs in the equine genome, we collected over 80 tissues from four healthy adult Thoroughbred horses as a part of the Functional Annotation of Animal Genome (FAANG) initiative. Detailed phenotyping and tissue collection protocols have been previously reported [38,39]. Nine prioritized tissues (lamina, liver, left lung, left ventricle of heart, longissimus muscle, skin, parietal cortex, testis, and ovary) were used to generate a diverse set of data that represent different aspects of gene expression and regulation. Additional RNA-seq data from 57 other tissues, generated from these same horses as a result of a community-driven effort, were used to compare our revised FAANG annotation with previous Ensembl and NCBI annotations. Here we present an integrated analysis of the equine FAANG dataset.

## Results

### Long-read data improved transcriptome annotation

Using Iso-seq data from nine prioritized tissues, we assembled a transcriptome with improved AS and 3' transcription termination site (TTS) annotation for Equcab3 [2]. This transcriptome contained 56,672 transcripts including 39,625 novel transcripts. A majority of these transcripts (51,639) were multi-exonic. Of these novel transcripts, 30,964 (78%) were AS isoforms with either novel combinations of known splice junctions (6,330) or novel splice junctions (24,634). Of the 17,407 known transcripts, 12,470 contain splice junctions fully matched to a reference transcript annotated in the Ensembl gene annotation [5,7] for EquCab3 (full-splice match, FSM). The majority (9,924 or 79.6%) of these transcripts extended reference annotation at the 3' end, with 4,232 transcripts having TTS more than 1 kb downstream of Ensembl annotated TTS (**Fig 1A**). The remaining 4,937 known transcripts lacked known splice junctions at either 5' or 3' end (incomplete-splice match, ISM), of which 2,395 extended the reference annotation at the 3' end (**Fig 1A**). At the transcription start sites (TSS), the majority (98.4%) of transcripts had higher RNA-seq coverage in 100bp windows downstream of TSS than upstream and 89.1% had at least twice coverage in 100bp windows downstream of TSS than upstream (**Fig 1B**).

The tissue-specific expression of these transcripts was quantified using short-read RNA-seq data from 57 tissues of the same animals, nine of which were the same tissues used to generate

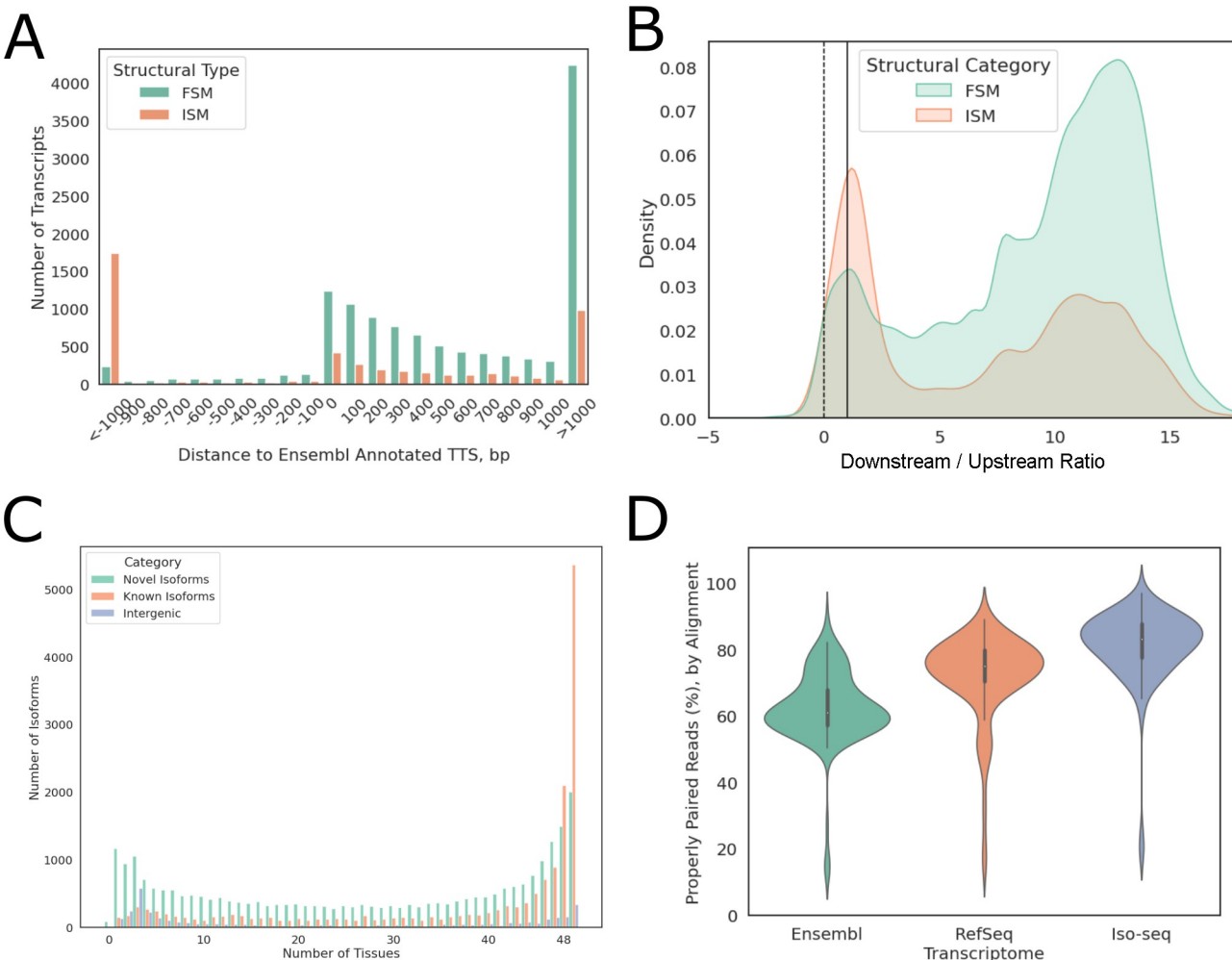

**Fig 1. Iso-seq transcriptome improved gene annotation.** (A) Distance between Iso-seq Ensembl annotated TTS, negative values indicate shorter 3' ends. Eg. -1000 indicates that Iso-seq annotated TTS is 1000 bp upstream of Ensembl annotated TTS. (B) $Log_2$ of 100 bp downstream as compared to upstream of TSS RNA-seq coverage. Positive ratios indicate higher coverage downstream of TSS. The dotted line indicates equal coverage up- and down-stream of TSS while the solid line indicates 100% higher coverage downstream of the TSS than upstream. (C) Distribution of known vs. novel transcripts detected in different numbers of tissues. (D) Distribution of mapping rates against Iso-seq transcriptome vs. Ensembl or RefSeq transcriptome across 57 RNA-seq samples. FSM: full-splice match; ISM: incomplete-splice match; TTS: transcription start site; TSS: transcription termination site.

Iso-seq data. Approximately 78% of known isoforms were expressed in at least half of the tissues sequenced, while novel isoforms of known genes and novel intergenic transcripts each showed a bimodal distribution, with 44.3% of novel isoforms and 56.8% of intergenic transcripts detected in less than half of the tissues (**Fig 1C**). We also noted that, on average, 61.4% (33.3%-70.9%) of multi-isoform genes expressed more than one isoform in any given tissue and had different dominant major isoforms (isoform with highest relative expression of a given gene), depending on the tissue type.

The completeness of this transcriptome annotation was assessed by aligning RNA-seq reads directly to transcriptome sequences. Compared to the Ensembl and RefSeq annotations, this Iso-seq transcriptome showed substantial improvement in mapping rates, as measured by percentage of properly paired reads, with a median mapping rate of 83.25% as compared to 61.10% and 75.15% when using Ensembl and RefSeq annotation, respectively (**Fig 1D**).

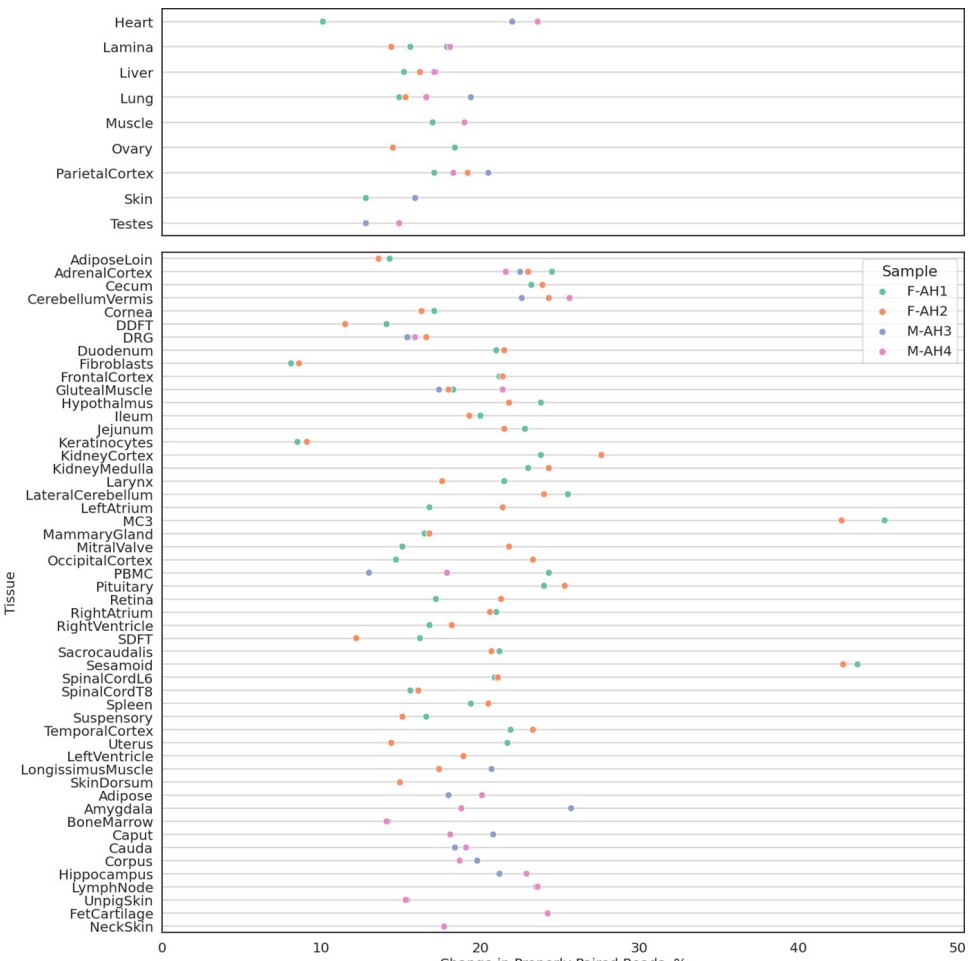

**Fig 2. Comparison of FAANG, RefSeq and Ensembl equine transcriptomes.** Changes in percentages of properly paired reads aligned to combined FAANG transcriptome when compared to Ensembl or RefSeq transcriptomes, whichever has higher percentage.

Since only nine tissues were used to construct this Iso-seq transcriptome, it was expected that many tissue-specific genes and transcripts will be missing. Indeed, several tissues had decreased mapping rates when aligned to the Iso-seq transcriptome as compared to Ensembl or RefSeq (S1 Fig), especially those with large stem cell populations. Therefore, this Iso-seq transcriptome was merged with Ensembl and RefSeq transcriptome to construct a more complete transcriptome annotation, termed the FAANG transcriptome. RNA-seq alignment data indicated that the FAANG transcriptome was substantially more complete than the existing annotations, with an average 19.5% (8–45%) increase in mapping rates across all sequenced tissues (Fig 2). The FAANG transcriptome consisted of 153,492 transcripts (of which 128,723 were multi-exonic) from 36,239 genes, with a gene-isoform ratio of 4.2.

## Tissue-specific open chromatin annotation

Chromatin accessibility was profiled from the same nine tissues (adipose, heart, lamina, liver, lung, ovary, testis, muscle, parietal cortex). Most libraries contained 60% to > 90% unique reads, with the exception of liver and cerebral cortex samples. Data from the female liver samples were generated from our previous study, where excessive mitochondria contamination

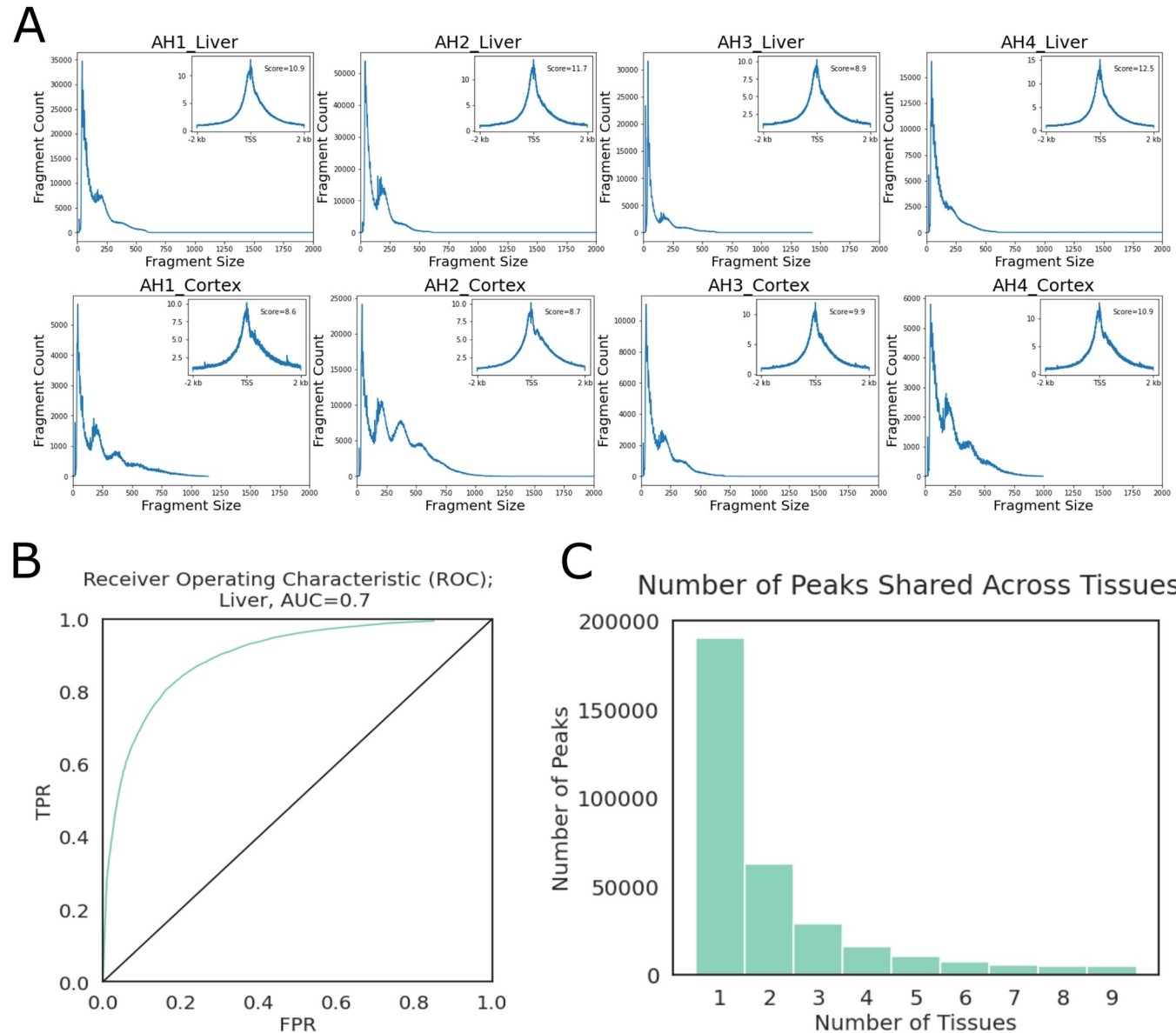

**Fig 3. ATAC-seq Quality Control.** (A) Fragment size distributions and TSS enrichment plots (upper right inset plots) of liver and cerebral cortex samples as examples. (B) ROC plot of liver peaks, as a representative example across tissues. TPR: true positive rate, FPR: false positive rate, AUC: area under curve. (C) Number of peaks that were observed in a given number of tissues.

led to lower library complexities and resequencing was performed to reach desired unique read counts [27]. After removing polymerase chain reaction (PCR) duplicate reads, all libraries contained less than 20% of mitochondria reads. Despite lower library complexities, both liver and cerebral cortex samples showed clear nucleosomal periodicities and high enrichment around TSS (**Fig 3A**).

Peaks were called by MACS3 [40,41] single-end BED mode using both ends of aligned fragments. Accuracies of peak calling were estimated using published histone ChIP-seq data from the same tissues [42]. Briefly, open chromatin peaks were intersected with "true positive" (TP, H3K4me1 or H3K4me3 peaks overlapping H3K27ac) and "true negative" (TN, H3K27me3 peaks) peak sets to calculate true positive rates (TPR) and false positive rates (FPR). Since testes

**Table 1. Open Chromatin Peak Metrics.** Number of peaks identified in each tissue before and after filtering as well as union and tissue-specific peaks.

| | Merged Raw Peak Count | Cutoff | TP | FP | TPR | FPR | Remaining Peak Count | Tissue Specific |
|---|---|---|---|---|---|---|---|---|
| **Adipose** | 941,236 | 67 | 10,595 | 2,008 | 0.59 | 0.25 | 77,655 | 22,884 |
| **Cortex** | 435,514 | 114 | 14,109 | 1,959 | 0.78 | 0.25 | 65,583 | 18,249 |
| **Heart** | 581,396 | 85 | 14,955 | 2,226 | 0.83 | 0.25 | 86,368 | 19,730 |
| **Lamina** | 722,387 | 89 | 9,423 | 1,765 | 0.48 | 0.25 | 63,136 | 21,805 |
| **Liver** | 557,874 | 76 | 16,078 | 2,815 | 0.87 | 0.25 | 95,048 | 31,460 |
| **Lung** | 522,294 | 103 | 13,672 | 1,957 | 0.76 | 0.25 | 59,024 | 8,447 |
| **Muscle** | 360,298 | 98 | 15,891 | 2,090 | 0.86 | 0.25 | 74,285 | 19,772 |
| **Ovary** | 463,426 | 109 | 12,583 | 1,767 | 0.64 | 0.25 | 66,726 | 16,588 |
| **Testis** | 520,160 | N/A | N/A | N/A | N/A | N/A | 78,164* | 31,880 |
| **Union** | | | | | 332,115 | | | |
| **Conserved** | | | | | 5,080 | | | |

* Testis peaks were filtered by score at 85th quantile since no histone peaks were available for this tissue

TP: number of true positive peaks, FP: number of false positive peaks; TPR: true positive rate; FPR: false positive rate; cutoff: cutoff score below which peaks were removed from final peak set; union: peaks found in any tissue, after iterative merging; ubiquitous: peaks found in all nine tissues

were not included in the histone ChIP-seq dataset from Kingsley et al. [42], testes' libraries were not evaluated at this step. Area under curve (AUC) values of at least 0.6 were achieved for all tissues evaluated (**Figs 3B and S2**). Cutoff scores were set at 25% FPR to filter a final set of peaks for each tissue, except testis. After filtering, the evaluated tissues had 59k-95k peaks remaining (**Table 1**). Testis and liver had the highest amounts of tissue-specific peaks (31,880 and 31,460, respectively), while lung had the lowest number of tissue-specific peaks (8,447). Only a very small number of peaks were conserved across examined tissues (**Fig 3C**).

These open chromatin peaks were annotated by their overlapping genic features as promoter-TSS (2 kb up- or down-stream of a TSS), exon, intron, TTS, and intergenic peaks. Open chromatin peaks across tissues were enriched in TSS, TTS, and exon regions (2.9-, 1.3-, and 1.3-fold enrichment, respectively). This enrichment was more apparent among peaks conserved across tissues (**Fig 4**). Gene ontology (GO) terms overrepresented in genes associated with these conserved peaks were all essential housekeeping biological processes such as TOR signaling and kinase activity (**S1 Table**). For each tissue, 11–22% peaks were located within promoter-TSS regions. However, the same pattern was not observed in tissue-specific peaks. Only 3–5% of tissue specific peaks were in the promoter-TSS regions, while substantially more peaks were located in intronic or intergenic regions (17–22% intronic, 21–34% intergenic

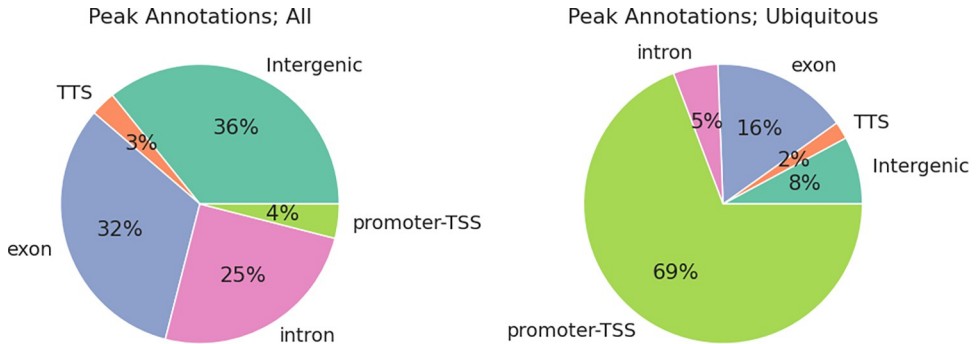

**Fig 4. Peak annotations.** Left: Composition of peaks by annotation in union peaks; Right: Composition of peaks by annotation in ubiquitous peaks.

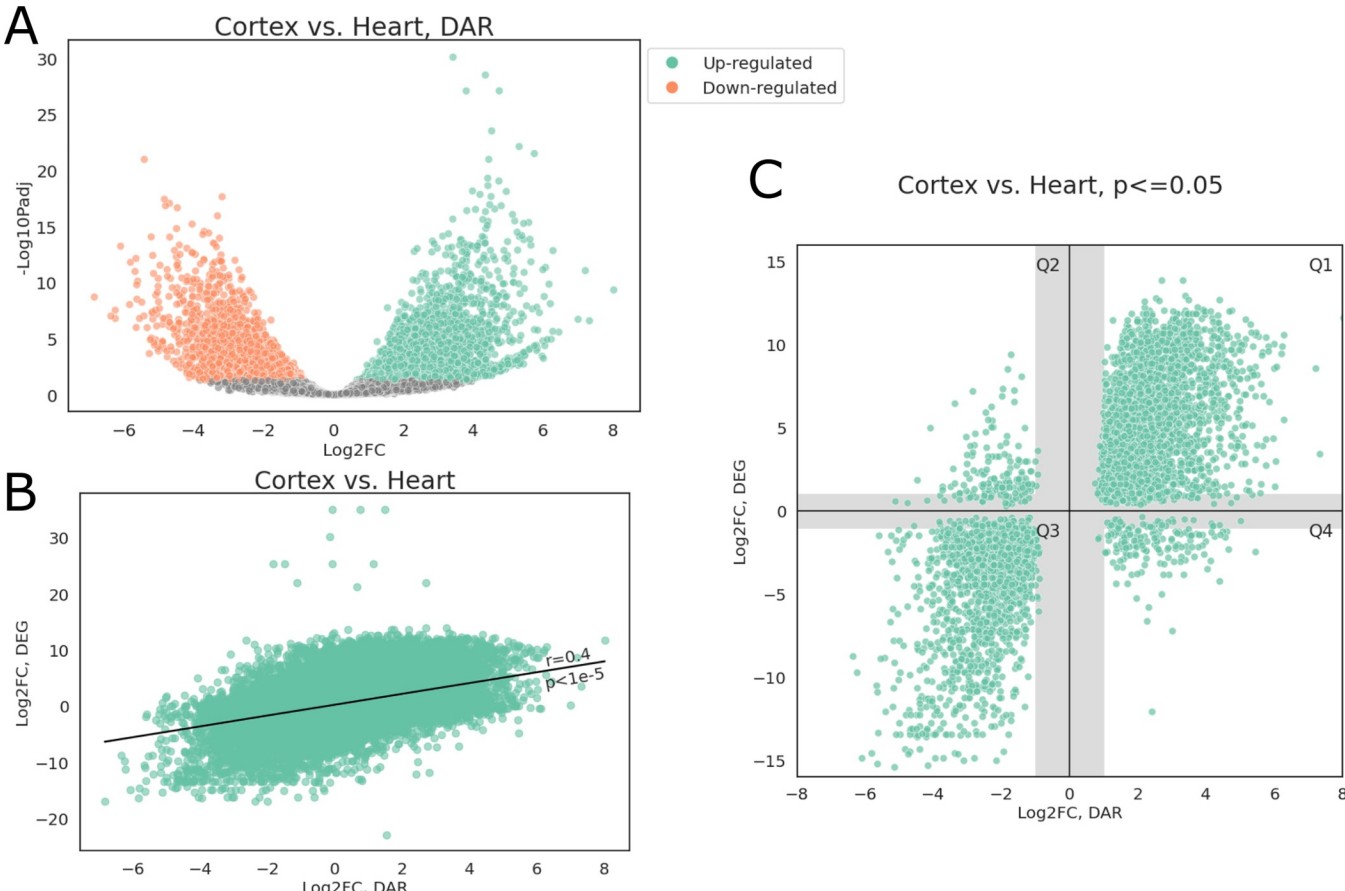

**Fig 5. Differential accessibility analysis.** (A) Volcano plot of open chromatin peaks; peaks with FDR adjusted p<0.05 and |log$_2$FC|>1 were colored by direction of accessibility change, positive log$_2$FC indicate greater accessibility in cerebral cortex. (B) Scatter plot of log$_2$FC from DEG and DAR analyses, Pearson correlation r = 0.4. (C) Scatter plot of log$_2$FC from DEG and DAR analyses, only those with both FDR adjusted p<0.05 were plotted; shaded areas indicate regions where either FC$_{DEG}$ or FC$_{DAR}$ was under 2-fold.

peaks across tissues; 23–26% intronic, 23–45% intergenic tissue-specific peaks). Motif analyses of these intergenic regions revealed a diverse range of TF binding sites, such as hepatocyte nuclear factor-4 alpha (HNF-4α) and estrogen-related receptor alpha (ERRα) binding sites in liver-specific intergenic open chromatin regions, myocyte enhancer factor-2 (MEF2) family TF binding sites in heart-specific open chromatin regions, and SRY-related HMG-box (SOX) family TF binding sites in cerebral cortex-specific open chromatin regions. (**S2 Table**).

Unsurprisingly, TSS accessibility showed significant correlations with their corresponding gene expression. **Fig 5** shows a representative differential accessibility and expression analysis of parietal cortex and heart tissues. Overall, approximately 16% of peaks showed differential accessibility (FDR adjusted p<0.05), with 25,144 peaks (7.6%) more accessible in cortex and 29,206 peaks (8.8%) more accessible in heart (**Fig 5A**). The log$_2$ fold-change (log$_2$FC) of differentially accessible regions (DAR) was significantly correlated with log$_2$FC of differentially expressed genes (DEG) in the same cortex and heart samples (one-sided Wald test, p<1 x 10$^{-5}$, Pearson correlation coefficient r = 0.4, **Fig 5B**). After selecting peak-gene pairs whose FDR adjusted p values from both DAR and DEG analyses were below 0.05, we observed that most genes were located in quadrants 1 and 3 (Q1 and Q3 respectively), showing concordant changes in promoter-TSS accessibility and gene expression (**Fig 5C**). GO enrichment analyses

showed that Q1 genes were primarily associated with neural activities while muscular and cardiac related GO terms were enriched among Q3 genes (**S3** and **S4** Tables). There were also 175 and 177 genes in Q2 and Q4, respectively. GO Terms related to synaptic activity were enriched in Q2 (**S5 Table**) while Q4 genes were overrepresented in actin-filament based processes.

## Cis-regulatory element annotation

Chromatin states were first identified using four major histone modifications (H3K4me1, H3K4me3, H3K27ac, H3K27me3) as well as CTCF binding from the same nine tissues. Overall, 14 unique states, corresponding to enhancer, promoter, and insulator states of various degrees of activities, as well as polycomb repressed state were identified (**Fig 6A**). Notably, the CTCF-bound active TSS state (state *4*), co-enriched with CTCF and active promoter marks (H3K4me3 and H3K27ac), was highly enriched around TSS, whereas the CTCF-less active TSS state (state *3*) was more enriched at approximately 500 bp up- and down-stream of TSS. Collectively, states with assayed epigenetic signals (states *1–13*) covered up to 20% of the genome, with the polycomb repressed state (state *13*) covering the largest portion of the genome across tissues, followed by enhancer states (states *6–10*, **Fig 6B**). While promoter states only accounted for 3–5% of the genome, or around 20% of all annotated states, they comprised over 50% of states annotated at TSS regions (**Fig 6B and 6C**).

To correlate gene expression with chromatin state annotation, companion RNA-seq data for each tissue from FAANG was used to quantify the equine FAANG transcriptome via

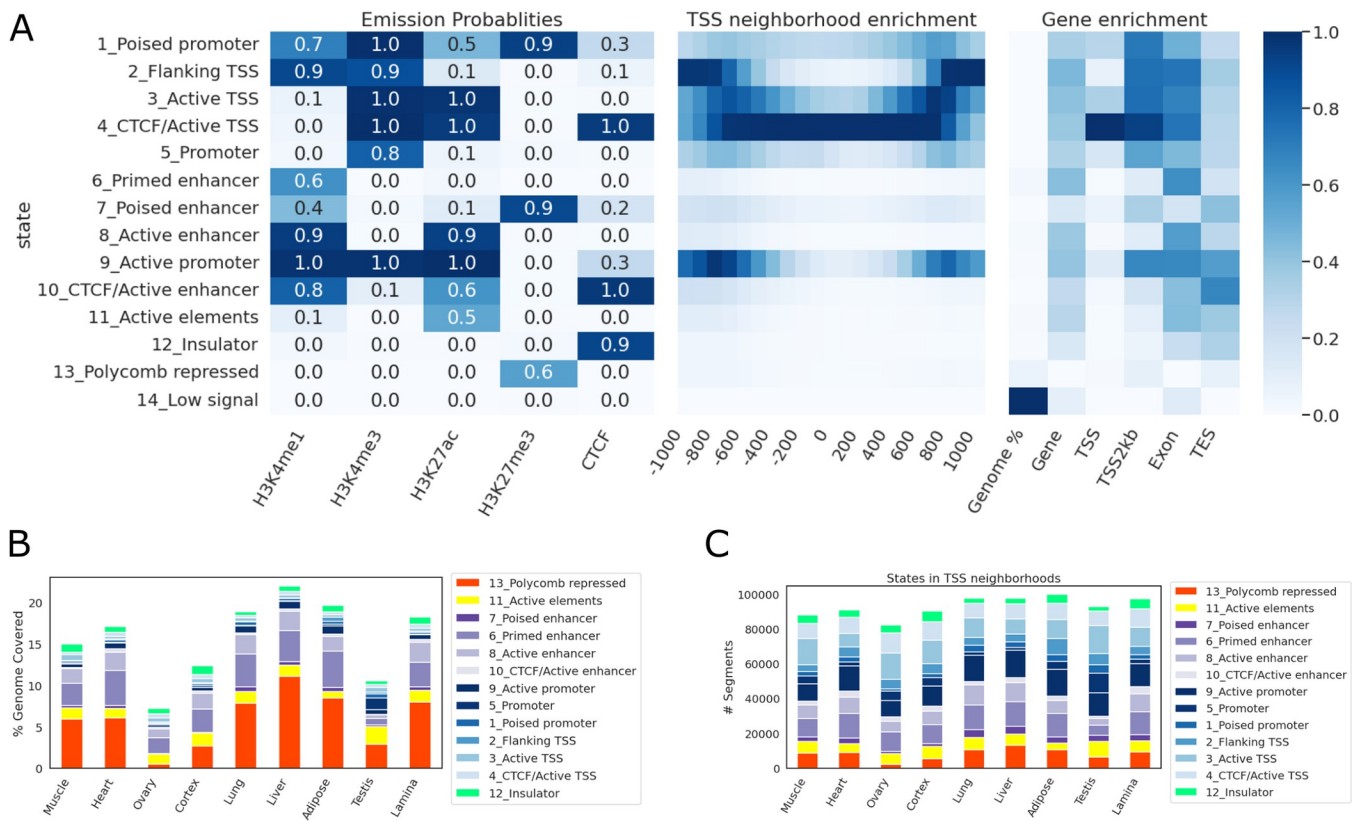

**Fig 6. Chromatin states.** (A) Emission probabilities, TSS neighborhood enrichment, and different genic features' enrichment at each state. (B) The percentage of genome covered by each state in each tissue. (C) The number of segments from each state in each tissue.

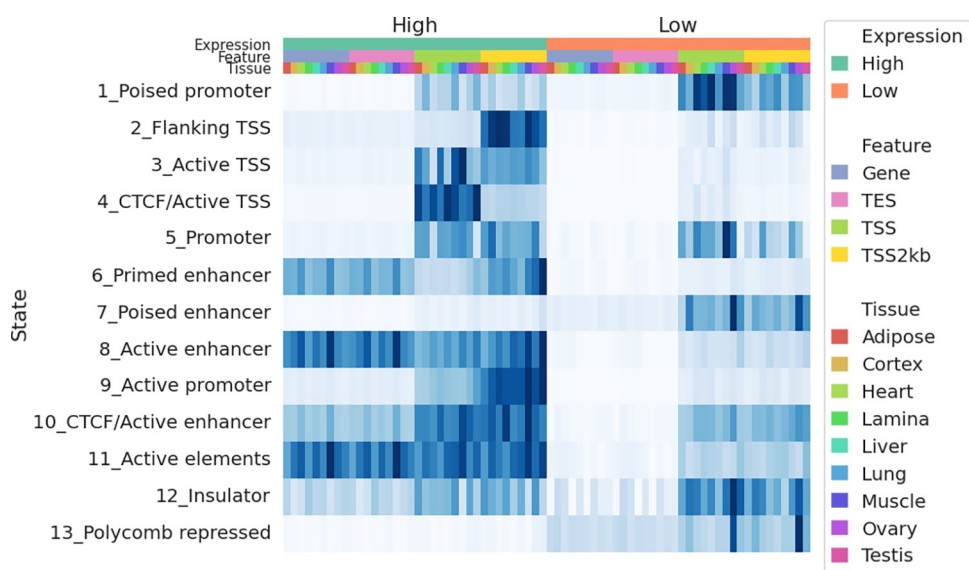

**Fig 7. State enrichment in tissue specific genes.** Heatmap of enrichment for each state around genes expressed and not expressed in each tissue. The top three color bars denote gene expression status, genic features, and tissues, respectively. Enrichment scores were normalized in each column.

quasi-mapping [43]. Transcript level quantification was then summarized to gene level using tximport [44]. For each tissue, genes were classified as high- or low-expression based on their aggregated transcripts per million (TPM) values (high: TPM ≥ 1; low: TPM < 1). The enrichment of each state was then estimated across gene bodies, in exonic regions, around TSS and transcription end sites (TES) across all nine tissues (**Fig 7**). CTCF bound active TSS state (state *4*) showed a 59.4-fold enrichment around TSS of highly expressed (TPM ≥ 1) genes, 7.7 times that of lowly expressed genes (TPM < 1). Similarly, active promoter state (state *9*) showed a 14.7-fold enrichment in promoter-TSS neighborhood (TSS2kb), 6.4 times that of lowly expressed genes. On the other hand, poised promoter and enhancer states (states *1* and *7*) were more enriched around TSS of lowly expressed genes (18.7- and 7.5-fold enrichment, respectively). Polycomb repressed states (state *13*) were absent around genes with high expression but enriched in low-expression genes while promoter state marked by a single H3K4me3 mark (state *5*) was observed at a similar level in both categories. Since this promoter state also showed the highest tissue-specificity (**S3 Fig**), we further examined its distribution among tissues. Most remarkably, testis harbored a substantially greater number of segments of State *5* than any other tissues (46,406 in testis compared to 5,235 in ovary, which was the next highest tissue) (**S4 Fig**). 61% of promoter state (state *5*) was found in testis and of those, 86% were specific to testis. Similarly, testis also contained the highest numbers of CTCF-less active TSS and poised promoter states (**S5 Fig**). While less pronounced, it also accounted for 44% of CTCF-less active TSS state (state *3*) and 54% of poised promoter state (state *1*).

To infer potential functions of open chromatin regions, especially those located in the intergenic regions, open chromatin peaks were annotated based on overlap chromatin state segments in each tissue. First, we examined overlap between each chromatin region and different open chromatin states across tissues (**Fig 8A**). There was an overall agreement in promoter-TSS assignment between open chromatin regions and chromatin state annotations: 92.9% of open chromatin peaks located in TSS-promoter regions overlap a TSS or promoter state (states *1–5, 9*). Additionally, open chromatin regions located in exonic and intergenic regions showed higher percentages of enhancer states (28.9% and 17.9%, respectively) (**Fig 8A**). Next, we

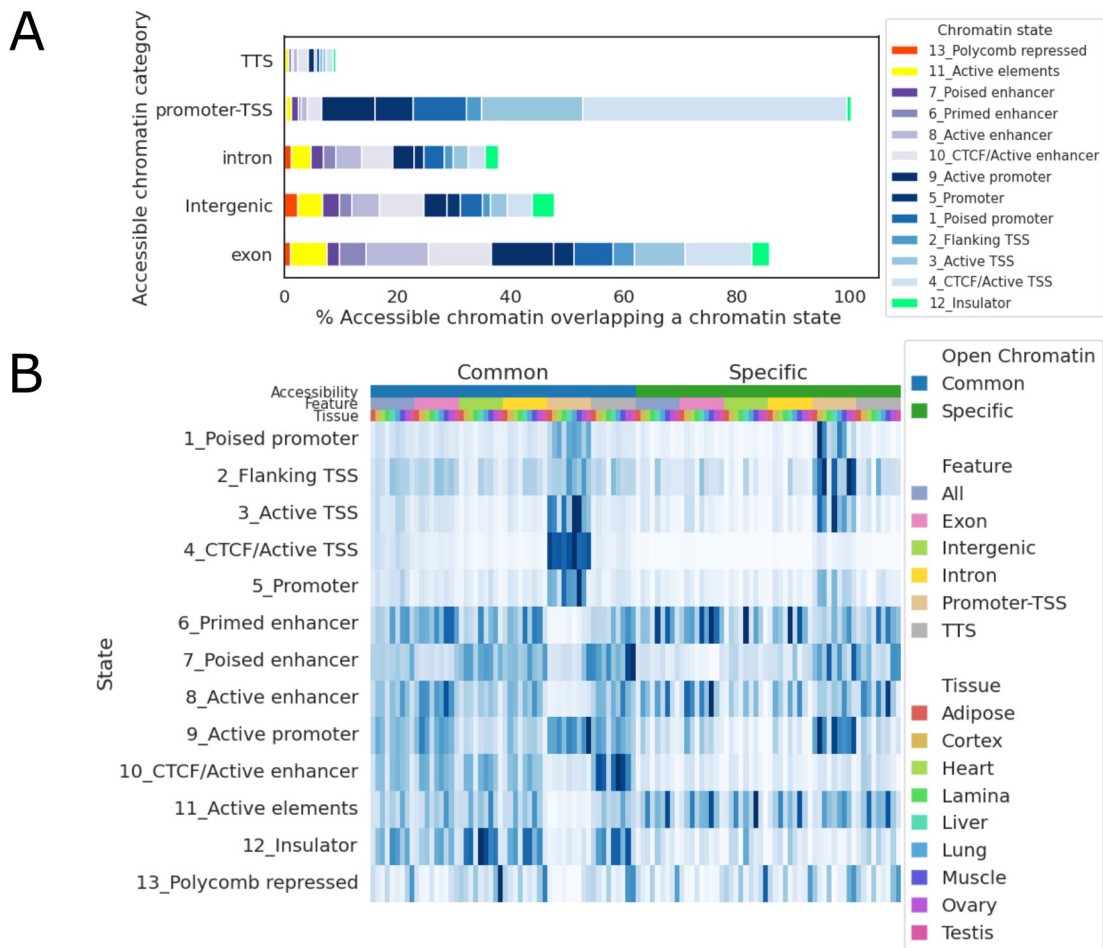

**Fig 8. Chromatin accessibility across states.** (A) Percentage of open chromatin peaks that overlap each chromatin state. (B) Heatmap of enrichment for each state around open chromatin peaks. The top three color bars denote shared or tissue-specific open chromatin, open chromatin annotation, and tissues, respectively. Enrichment scores were normalized in each column.

compared chromatin state enrichment among shared and tissue-specific open chromatin regions (**Fig 8B**). An open chromatin region was annotated as specific if it was found in only one tissue. CTCF bound active TSS state (state *4*) was highly enriched in common accessible chromatin regions, especially those annotated as promoter-TSS regions (121-fold enrichment), but much less so in tissue-specific accessible chromatin regions (12-fold enrichment). Similarly, CTCF bound enhancer state (state *10*) was also highly enriched in common accessible chromatin regions outside of promoter-TSS neighborhoods (15.3- to 31.5-fold enrichment), and less so in tissue-specific accessible chromatin regions (3.9- to 7.6-fold enrichment).

Since many enhancers interact with genes other than their nearest neighbors [45], in order to predict target genes CREs in absence of chromatin interaction data, we predicted chromatin loops and correlated H3K27ac signal (data originally generated and reported by Kingsley et al. [42]) of CREs with expression of genes resided within the same chromatin loops. First, chromatin loops were predicted using CTCF ChIP-seq data, as described by Oti *et al.* [46]. Overall, we identified 10-14k CTCF-mediated loops per tissue, with testis being the only exception, having only 6,146 CTCF-mediated loops. In all tissues, including testis, these predicted loops covered 80–85% of the genome. Since we only had at most 4 biological replicates per tissue (two for ovary and testis samples), which was not enough samples to reliably estimate

Spearman correlation coefficients [47], we opted for a pan-tissue approach. Tissue-wise chromatin loops were merged across tissues to form a catalog of pan-tissue CTCF-mediated chromatin loops, enabling estimation of correlation across 9 tissues and 4 biological replicates. This catalog contained 4,556 non-overlapping loops, covering 94.0% of the equine genome.

As demonstrated by Kern *et al.* [48], H3K27ac intensity of a CRE is most tightly correlated with its target gene's expression level. Therefore, we correlated H3K27ac ChIP-seq read counts and RNA-seq read counts of each CRE-gene pair that resided within a same predicted chromatin loop. After adjusting for multiple testing using Benjamini-Hochberg procedure to control the false discovery rate at 5%, a total of 84,613 CRE-gene pairs remained as candidates. These CREs were then annotated as genic, intergenic, or TSS-proximal based on their relative proximities. A majority of these candidate pairs had their CREs outside of the gene bodies or TSS-proximal regions (intergenic CREs, 66,051) while only a small portion of them had promoter-like relationship (CREs in the TSS-proximal regions, 8,225). Intergenic CREs were found at varying distances to TSS, with a median distance of 200 Kb and 79% of CREs being within 1 Mb their target TSS. We also observed more CREs (75%) located downstream of their target genes (**S6 Fig**).

To provide the equine community with an integrated, openly access FAANG dataset, we developed a UCSC track hub (https://genome.ucsc.edu/s/cjfinno/equCab3) to host all currently published equine FAANG datasets. All features discussed above can be found in this track hub.

## Discussion

In this study, we detailed the efforts to create an integrated annotation for the horse genome that will aid in deeper understanding of gene expression and regulation across tissues in the horse. Utilizing the rich tissue repository from the equine FAANG project, we improved the equine transcriptome with 39,625 novel transcripts and identified 84,613 candidate CRE-gene pairs, 78.1% of which were intergenic CREs. We anticipate these new resources will play a vital role to understanding how genetic variation in the horse contributes to equine biology and health.

First, we outlined an expansive transcriptome annotation for the horse. The previous efforts to annotate the horse genome were limited by the number of tissue types available and sequencing lengths available at that time [49–51]. Specifically, Hestand et al. sequenced 43 different equine tissues in one pool on an Illumina HiSeq 2000, both single- and paired-end at 75 bp on 4 lanes each [50]. While that study included more diverse tissue types than the current FAANG transcriptome, the pooling approach employed in that study limited discovery of rare novel and tissue-specific isoforms. The non-stranded protocol employed in that study also rendered it impossible to identify antisense transcripts. Mansour et al. compared sequences of 8 tissue samples from 59 individuals using short-read RNA-seq libraries from several studies (80–125 bp, single- and paired-end, stranded and unstranded) and identified 36,876 genes with 76,125 isoforms [51]. Due to the limitation of short-read sequencing technology in both the Hestand and Mansour studies, an aggressive filter was necessary to remove mono-exonic transcripts that were not evolutionarily conserved, a common strategy in short-read based transcriptome assemblies [4,5]. This unfortunately would also remove many small noncoding RNAs. Based on recent advances in long isoform sequencing, our approach centered around high-quality full-length reads from Iso-seq and used abundant RNA-seq data to validate splice junction, TSS, and TTS annotation. As a result, novel alternate-spliced isoforms as well as extended 5' and 3' transcribed regions were identified using long-read Iso-seq, validated by abundant short-read mRNA-seq. This combined approach expanded the equine transcriptome

to 153,492 transcripts (of which 128,723 are multi-exonic) from 36,239 genes, with a gene-iso-form ratio of 4.2 and an average 19.5% (8–45%) improvement in completeness compared to Ensembl and RefSeq transcriptomes across all sequenced FAANG tissues. The newly discovered genes and isoforms could help identify important coding or regulatory variants in the horse.

Despite these improvements, the present Iso-seq transcriptome was unable to accurately define TSS due to a lack of 5' captured reads. While an aggressive approach was taken to ensure 5' completeness by collapsing transcripts that only differ at 5' ends, a small portion of transcripts were still determined to be potentially 5' incomplete. In addition, this approach may hinder the discovery of alternative TSS. Furthermore, small RNAs with non-polyadenylated tails are missing from the poly-A captured cDNA libraries used for both Iso-seq and RNA-seq. Assays targeting non-polyadenylated RNAs, such as small RNA sequencing and techniques capturing 5' capped transcripts like CAGE-seq are necessary to complement this Iso-seq transcriptome to fully capture the transcriptional landscape in the horse genome. Further, while we demonstrate improved completeness of the FAANG equine transcriptome, only 9 tissues were utilized to construct it, and many rare or tissue-specific transcripts are likely to be missing, especially stem-cell-specific or embryonically specific transcripts. Indeed, short read sequencing data from bone marrow was the only tissue that showed a drastic decrease in mapping rate when compared to RefSeq or Ensembl transcriptomes, suggesting specific isoforms from this tissue are missing in the new transcriptome. In addition, since mare and stallion tissues were prepared at two different laboratories, despite using same protocols, we could not distinguish any sex-specific expression from batch effects during RNA-seq library construction. Lastly, since RNA-seq data was only used to validate and quantify transcripts identified in Iso-seq data, our approach heavily relied on Iso-seq data being complete. This unfortunately was not true as it was evident in **Figs 2** and **S1** that a substantial amount of reads from RNA-seq failed to map to Iso-seq transcriptome. A closer examination of RNA-seq data in conjunction with the Iso-seq data could further refine the FAANG transcriptome.

With an improved transcriptome annotation, we set out to identify other non-transcribed or lowly transcribed regulatory regions in the horse genome. The first step was to identify regions of the horse genome that were accessible to transcription factors, which can then serve as proxies to identifying important regulatory regions. Using ATAC-seq, we identified 332,115 regions with open chromatin genome wide across tissues, with 59,024–95,048 peaks identified in each tissue. We showed that these open regions were enriched with known TF binding sites, further supporting their potential functional roles in gene regulation.

We observed that, while promoter-TSS regions were highly enriched in open chromatin peaks, especially in peaks found in all tissues, they were conspicuously absent from tissue-specific peaks. This echoed the findings from Halstead *et al.* [52], which showed very low numbers of TSS-related peaks in species-specific open chromatin regions. This would corroborate recent findings that enhancers, not promoters, are the main drivers of tissue-specific transcription [53], which would be classified as intergenic in both our study as well as in Halstead *et al.*, as all species interrogated in these studies lacked enhancer annotation.

Combining our ATAC-seq data with previously reported mRNA-seq data from the same tissue and samples, we showed a strong correlation between differential accessibility of a functional element and differential expression of the corresponding gene. When FDR for both DAR and DEG was controlled at 5%, we observed concordant patterns between the gene expression level and accessibility of promoter-TSS region. Interestingly, a small number of genes (352 out of 4,566, 7.8%) also showed discordant patterns between gene expression and promoter-TSS accessibility. It should be noted that, while we took a similar analytical approach for DEG and DAR, ATAC-seq and RNA-seq signals reflect fundamentally different regulatory

features. RNA-seq captures total transcriptional activities in a population of cells, where a small population of cells with extremely high transcription of a given gene can dominate RNA-seq signals for the entire population. On the other hand, ATAC-seq captures the proportion of cells whose DNA is accessible at a given locus. Thus, if a small population of cells have substantially high expression of a given gene, while the remaining cells do not express this gene nor is it accessible, the gene would be upregulated in the RNA-seq dataset but not identified as open in an ATAC-seq dataset, which could explain the discordant patterns we observed in Q2 and Q4 genes. Additionally, the presence of silencers in CREs or the time difference from CRE activation to change in mRNA abundance may also explain the observed discordance. To further dissect the fine regulatory landscape of these genes, single-cell based approaches are advisable for both RNA quantification and chromatin accessibility assessment to best match representing cell types across assays.

In all nine tissues, 21–34% of peaks were located in the intergenic regions while 23–45% of tissue-specific peaks were in intergenic regions. Motif analyses in this study identified common TF binding sites in many of these intergenic open chromatin regions. For example, over 41% (4,112) of the liver-specific intergenic open chromatin regions contained binding sites for ERRα, a TF known as a central regulator of energy metabolism [54]. Similarly, binding sites for various SOX family TFs were detected in 15–32% of cerebral cortex-specific intergenic open chromatin regions. The SOX family TFs have been shown to be important regulators for neural differentiation and adult neurogenesis [55].

It is likely that many of intergenic open chromatin regions have important regulatory functions. To elucidate their functional roles, we identified potential regulatory states genome wide across tissues using signals from biochemical assays (histone modifications and CTCF ChIP-seq) and correlated them with open chromatin peaks. We identified 14 unique chromatin states using data from four major histone marks and CTCF binding assays. These chromatin states were identified in each of the nine tissue types, covering 7–21% of the genome, representing major CREs. Overall, the chromatin state annotation correlated well with chromatin accessibility in the same tissues and provided additional information regarding potential function of REs in these tissues. These annotated REs will be an invaluable addition to the equine reference genome assembly. The similar annotation provided by ENCODE has led to discoveries of many regulatory variants in various diseases [20,22,56]. We anticipate this catalog of REs will prove instrumental in evaluating complex genetic traits and disease in the horse.

In developing these unique chromatin states, we noted a particular difference in chromatin state annotation for shared and tissue-specific open chromatin peaks. CTCF bound promoter and enhancers were highly enriched in shared open chromatin regions but less so in tissue-specific regions. CTCF is a chromatin regulator that facilitates formation of chromatin loops. It has been suggested that a subset of CTCF binding sites is constitutively bound and critical to well-regulated gene expression [57] and that CTCF binding at proximal promoters promotes distal enhancer-promoter interaction, which is essential to the activation of many genes across a diverse range of tissues [58]. Our results suggest that these CTCF-mediated promoter-enhancer interactions play a large role in genes expressed across multiple tissues, rather than tissue-specific genes. This aligns with other findings which suggest that CTCF patterns are established early in embryogenesis [59]. Taking advantage of the extensive research surrounding the relationship between CTCF binding and 3-dimensional chromatin structures (TADs), we used our CTCF ChIP-seq data to predict chromatin loops and were therefore able to predict 84,613 candidate CRE-gene interactions across tissues. This dataset should dramatically improve our ability to both identify important regulatory variants and predict their target genes and gene networks.

Work from this study has opened doors for further exploration. For example, the discordant relationships between differentially accessible regions (DARs) and differentially expressed

genes (DEGs) in brain and heart tissues suggest substantial sub-population differences within each of these tissues. Future studies utilizing single-cell-based technologies could help unravel such differences and identify cell-type-defining genes and CREs. Additionally, we observed substantial differences between testes and all other tissues from both the ATAC-seq and ChIP-seq data. This difference could be a result of significant spermatozoa population in our testis samples, or it could be related to the unique transcriptional landscape of testis. Future research should focus on separating mature spermatozoa with spermatogonium and other cell types in testis to further refine the regulatory landscape of this tissue. Overall, we presented an integrated repository of equine FAANG data, encompassing both transcriptional and regulatory features that are now freely available to the equine community. We anticipate this resource to be integral to future equine research.

## Methods

### RNA extraction and sequencing

From the outset of the equine FAANG initiative, researchers were invited to "adopt" tissues of interest. This involved sponsorship of the sequencing costs for two biological replicates (2 male or 2 female) of the "adopted" tissue. Under this Adopt-A-Tissue model, along with the eight prioritized tissues funded by both the USDA National Institute of Food and Agriculture and the Grayson Jockey Club Foundation, the equine community collectively generated short-read mRNA-seq data from over forty tissues. All RNA extractions for mRNA-seq were performed at two locations (female samples at UC Davis, male samples at University of Nebraska-Lincoln). Briefly, tissue aliquots were homogenized using Biopulverisor and Genogrinder in TRIzol reagent (ThermoFisher Scientific, Waltham MA). RNA was isolated and purified using RNeasy Plus Mini/Micro columns (Qiagen, Germantown, MD) or Direct-zol RNA Miniprep Plus (Zymo Research, Irvine, CA). A detailed protocol can be found in **S1 and S2 Texts**. For the female tissues, cDNA libraries were prepared with Illumina TruSeq Stranded kit and sequenced at the University of Minnesota sequencing core facility on an Illumina HiSeq 2500 using 125 bp paired-end reads. Male samples went through similar library preparation before 150 bp paired-end sequencing at Admera Health (South Plainfield, NJ) on an Illumina NovaSeq.

Nine tissues (lamina, liver, left lung, left ventricle of heart, longissimus muscle, skin, parietal cortex, testis, and ovary) from the FAANG biobank [38,39] were selected for Iso-seq to represent a wide range of biological functions and therefore, gene expression. RNA for Iso-seq was extracted separately from the same tissues as mRNA-seq using the same protocol. All tissues were processed in one batch for Iso-seq, except for parietal cortex, which was processed in a separate batch as a pilot study. One sample per sex per tissue was selected for sequencing based on sample availability and RNA integrity numbers (RINs selected > 7). cDNA libraries were prepared and sequenced at UC Berkely QB3 Genomics core facility. Two libraries were randomly pooled and sequenced on a single SMRT cell on PacBio Sequel II.

### Transcriptome assembly

Pooled subreads were first demultiplexed using Lima (https://lima.how/). Circular consensus reads (ccs) were then constructed from demultiplexed subreads using PacBio Ccs program (https://ccs.how/). PolyA tails were trimmed from ccs reads using Isoseq3 (https://github.com/PacificBiosciences/IsoSeq). This step also removes concatemers and any reads lacking at least 20 bp of polyA tails. Redundant reads were then clustered based on pair-wise alignment using Isoseq3. Clustered transcripts were aligned to the reference genome EquCab3 [2] using minimap2 [60] without reference annotation as guide. Collapsed transcripts were filtered if they

were not supported by at least two full length reads. Filtered transcripts from each sample were then merged into a single transcriptome using Cupcake (https://github.com/Magdoll/cDNA_Cupcake/) and further filtered to retain only those detected in more than one sample. The merged total transcriptome was again aligned to the reference genome and collapsed to remove redundant transcripts. Potential 5' degraded transcripts were also removed by collapsing transcripts that had identical 3' ends and only differed at 5' ends. SQANTI3 [61] was then used to classify and annotate the transcriptome against the RefSeq transcriptome as reference. Finally, the total transcriptome was filtered to remove nonsense-mediated decay transcripts, transcripts with a splice junction not covered by short-read RNA-seq data, and transcripts without short-read coverage support to generate the final FAANG equine transcriptome (5,546, 7,262, and 12,900 transcripts were removed by each filter, respectively). To detect potential intra-primed transcripts, the percentage of adenines in a 20 bp window immediately downstream of the annotated TTS was calculated for every Iso-seq transcript. Transcripts with 80% or more adenines (i.e., allowing for 4 mismatches with poly-T oligonucleotides) in a 20 bp window downstream of annotated TTS were designated as potential intra-priming candidates. Data processing, visualization, and statistical analyses were performed using pandas [62], matplotlib [63], seaborn [64], scipy [65], and scikit-learn [66]. Detailed program versions, commands, parameters, and code can be obtained from https://github.com/FinnoLab/FAANG_IsoSeq.

## RNA-seq analysis

Short-read RNA-seq data were trimmed to remove adapters and low-quality reads using trim-galore (https://www.bioinformatics.babraham.ac.uk/projects/trim_galore/) and Cutadapt [67]. Read qualities were inspected using fastQC [68] and multiQC [69]. Trimmed reads were aligned to equCab3.0 using STAR aligner [70] with standard parameters (with—outSAM-strandField intronMotif—outSAMattrIHstart 0). PCR duplicates were marked using sambamba [71]. Mapping rates, qualities, and fragment lengths were assessed with SAMTools [72] and deepTools [73]. Aligned reads were used to assess completeness of transcriptomes using deepTools. BWA MEM [74] was used to align the RNA-seq reads directly to transcriptomes and SAMTools was used to calculate the percentages of properly paired reads from the transcriptome alignment. Due to the presence of alternatively spliced isoforms in transcriptomes, multiple-alignment reads were not removed. Salmon [43] was used to quantify the transcripts using RNA-seq data. Transcript level TPM values were summarized to gene level using tximport [44]. For the chromatin state enrichment analyses, genes were designated "active" if its aggregated TPM was at least 1 in a tissue. TSS, promoter-TSS neighborhood (TSS±2kb), exon, intron, and TTS coordinates were determined for each gene based on the FAANG transcriptome. Detailed program versions, commands, parameters, and code can be obtained from https://github.com/FinnoLab/FAANG_IsoSeq.

## ATAC-seq analysis

ATAC-seq data from the 9 tissues (adipose, lamina, liver, left lung, left ventricle of heart, longissimus muscle, parietal cortex, testis, and ovary) of two sexes collected from the equine FAANG biobank [38,39] were generated according to Peng et al. [27] Libraries were sequenced in 50 bp paired-end mode (PE50) on Illumina NovaSeq 6000. Reads were aligned to EquCab3.0 using BWA MEM with default parameters. Alignments were filtered to remove fragments that mapped to mitochondria genome, were discordantly mapped, PCR duplicates, or mapped to multiple loci using SAMTools. Remaining reads were shifted +4/-5 bp on plus/minus strand, respectively, to account for the 9 bp insertion introduced by Tn5 transposase

[24] using deepTools. Both forward and reverse reads of the final fragments were converted to bed format using bedtools [75] and peaks were called and refined using MACS3 [40,41] (-f BED -p 0.01—shift -75—extsize 150—nomodel—call-summits—nolambda—keep-dup all). After peak calling, we extracted summits from called peaks, and extended them on both sides by 250 bp, resulting in a set of 501 bp fixed length peaks. These peaks were then sorted by their score and non-overlapping, most significant peaks were retained, as described in Grandi *et al.* [76]. The same procedure was employed to subsequently merge biological replicates and then all tissue peak sets to generate a union set of peaks. A count matrix was constructed for the union peak set containing number of transposition events per peak per sample. This count matrix was used for differential accessibility analyses using DESeq2 [77]. The union peak set was then intersected with each tissue peak set to determine if a peak was present in each tissue. Peaks only identified in one tissue type were denoted "unique" peaks while those identified in all nine tissues were denoted as "conserved". Detailed program versions, commands, parameters, and code can be obtained from https://github.com/FinnoLab/atac-seq.

## Differential accessibility and expression analyses

DAR and DEG were analyzed with similar approaches. First, a matrix of raw counts was constructed containing number of transposition events (DAR) or RNA-seq reads (DEG) for each union open chromatin peaks (DAR) or gene (DEG). The raw counts were then normalized and fitted to a negative binomial generalized linear model using DESeq2 (1.30.1) [77] package with default parameters. Wald tests were applied to obtain p-values for each region (DAR) or gene (DEG) and Benjamini-Hochberg procedure were used to control false discovery rate at 5%.

## ROC analyses

For each set of peaks merged by tissues, false positive rates (FPR), true positive rates (TPR), and precision were calculated using published Histone ChIP-seq peaks from Kingsley *et al.* [42]:

First, a set of "real positive" (RP) peaks were collected by merging H3K4me1 and H3K4me3 peaks and intersecting the merged peaks with H3K27ac peaks from each tissue. A set of "real negative" (RN) peaks were collected from H3K27me3 peaks from each tissue. Subsequently, each set of ATAC-seq peaks were intersected with RP and RN peaks, and the number of intersections were recorded as "true positive" (TP) and "false positive" (FP). TPR, FPR, and precision were then calculated as follows:

$$TPR = \frac{n_{TP}}{n_{RP}}$$

$$FPR = \frac{n_{FP}}{n_{RN}}$$

$$Precision = \frac{n_{TP}}{n_{TP} + n_{FP}}$$

## Motif and gene ontology analyses

Motifs were analyzed using HOMER [78] (-size 250) with custom genome built from EquCab3.0 assembly and FAANG transcriptome annotation. GO enrichment analyses were performed using PANTHER [79] with default parameters.

## Chromatin state discovery

ChIP-seq data for histone modifications were obtained from previously published studies [42,80]. Additionally, ChIP-seq for CTCF was performed for the same nine frozen tissue samples at Diagenode *Inc.* (Belgium). Briefly, CTCF ChIP libraries were sequenced at 50bp single- and paired-end (female and male samples, respectively). Reads were aligned to EquCab3.0 using BWA MEM with default parameters. Aligned reads were subsequently filtered to remove low-quality mapping, PCR duplicates, and mitochondria reads using SAMTools. BAM files for all five marks were binarized using ChromHMM [81] BinarizeBam (-b 100 -n 140 -p 0.00001) and several models with different numbers of states were trained on binarized data using LearnModel function (-b 100). A model with 14 states was selected because it had the minimum number of states with strong correlation to all states identified in other models.

## Chromatin loop prediction and CRE-gene interaction analyses

To obtain a pan-tissue set of chromatin loops in absence of chromatin interaction data, CTCF ChIP-seq data was used to predict chromatin loops in each tissue, using the algorithm proposed by Oti *et al.*[46]. Briefly, CTCF ChIP-seq peaks were identified using MACS3 [40] (-q 0.05 -f BAM -g 2365156725—keep-dup auto). CTCF motifs were then extracted from these peaks using FIMO function from MEME [82] (—motif MA1930.1—parse-genomic-coord). Then in each tissue, each chromosome was scanned continuously, and a loop was recorded by detecting a pair of parallel and anti-parallel CTCF motifs. The predicted CTCF loops across tissues were then combined by merging overlapping loops using bedtools [75] merge function.

To obtain a list of CREs, neighboring active chromatin states (states *2–5* and *8–11)* were merged and annotated by their proximity to a known gene (genic, intergenic, and TSS-promoter) as well as their closest genes. These CREs were then quantified by number of overlapping reads from their corresponding H3K27ac ChIP-seq data. Reads counts were normalized by a scaling factor calculated using weighted trimmed mean (TMM) method. Normalized RNA-seq read counts were obtained from previous analysis (see Methods - Differential Accessibility and Expression Analyses). Spearman correlation between H3K27ac read count in each CRE and RNA-seq read count in each gene was calculated using spearmanr function from SciPy [65]. P-values were adjusted using Benjamini-Hochberg procedure and candidate CRE-gene pairs were filtered at 5% false discovery rate.

## Enrichment analysis

Enrichment of each state in genes and open chromatin regions was calculated using the following formula:

$$\frac{\frac{N_{Ann \bigcap State}}{N_{Ann}}}{\frac{N_{state}}{N_{genome}}}$$

where $N_{Ann}$ is the number of bases in a particular annotation (gene, exon, TSS, open chromatin peaks, etc) and $N_{state}$ is the number bases in each state. $N_{Ann \bigcap State}$ refers to the number of bases that are in both a particular state and annotation. $N_{genome}$ is the total size of the reference genome.

## Supporting information

**S1 Fig. Comparison of FAANG, RefSeq and Ensembl equine transcriptomes.** Changes in percentages of properly paired reads aligned to combined Iso-seq transcriptome when

compared to Ensembl or RefSeq transcriptomes, whichever has higher percentage.
(TIF)

**S2 Fig. ROC plots of ATAC-seq peaks.** Receiver Operating Characteristics (ROC) of eight tissues whose ATAC-seq peaks were validated by Histone ChIP-seq data
(TIF)

**S3 Fig. Tissue-specificity of states.** The proportion of segments from each state that were identified in different numbers of tissues.
(TIF)

**S4 Fig. Promoter state shared across tissues.** Intersection plot showing number of segments annotated as promoter state (state *5*) unique to each tissue and shared across tissues. Top: bar plot indicates sizes of each intersection; Bottom right: each column denotes a unique set of peaks where filled dots indicat that peaks in this set were found in the corresponding tissue; Bottom left: bar plot indicates number of segments annotated as promoter state (state *5*) in each tissue.
(TIF)

**S5 Fig. CTCF-less active TSS and poised promoter states shared across tissues.** Intersection plots showing number of segments annotated as (A) CTCF-less active TSS state (state *3*) and (B) poised promoter state (state *1*) unique to each tissue and shared across tissues. Top: bar plot indicates sizes of each intersection; Bottom right: each column denotes a unique set of peaks where filled dots indicate that peaks in this intersection were found in the corresponding tissue; Bottom left: bar plot indicates number of segments annotated as (A) CTCF-less active TSS state (state *3*) or (B) poised promoter state (state *1*) in each tissue.
(TIF)

**S6 Fig. Distance from intergenic RE to target genes' TSS.** Density plot of distances from intergenic REs to their target genes' TSS. Negative distance denotes RE being upstream of target TSS. Median absolute distance: 200 Kb.
(TIF)

**S1 Table. Gene ontology of peaks identified across all 9 tissues.** GO enrichment analysis of peaks conserved across all nine tissues.
(XLSX)

**S2 Table. De novo motif discoveries in tissue specific intergenic regions.** Top known motifs in each tissue, filtered by FDR q≤0.05.
(XLSX)

**S3 Table. Gene ontology of Q1 genes.** GO enrichment analysis of Q1 genes upregulated in both DEG and DAR analyses of cerebral cortex vs. heart.
(XLSX)

**S4 Table. Gene ontology of Q3 genes.** GO enrichment analysis of Q1 genes downregulated in both DEG and DAR analyses of cerebral cortex vs. heart.
(XLSX)

**S5 Table. Gene ontology of Q2 genes.** GO enrichment analysis of Q1 genes upregulated in DEG but downregulated in DAR analyses of cerebral cortex vs. heart.
(XLSX)

**S1 Text. Detailed protocol for RNA Isolation using a column and on-column DNA digestion.**
(DOCX)

**S2 Text. Detailed protocol for Finno modifications of the RNeasy Lipid Tissue Kit (Qiagen) for sesamoid bone.**
(DOCX)

**S1 Data. Gtf file for the merged FAANG-refseq-ensembl annotated transcripts in EquCab3.0.** This is a tab-delimited text formatted file can be uploaded to the Integrated Genome Viewer (IGV; https://software.broadinstitute.org/software/igv/ or UCSC https://genome.ucsc.edu).
(GZ)

## Author Contributions

**Conceptualization:** Sichong Peng, Jessica L. Petersen, Rebecca R. Bellone, Ted Kalbfleisch, Carrie J. Finno.

**Data curation:** Sichong Peng, Anna R. Dahlgren, Callum G. Donnelly, Erin N. Hales, Jessica L. Petersen, Rebecca R. Bellone, Ted Kalbfleisch, Carrie J. Finno.

**Formal analysis:** Sichong Peng, Erin N. Hales, Jessica L. Petersen, Rebecca R. Bellone, Ted Kalbfleisch, Carrie J. Finno.

**Funding acquisition:** Jessica L. Petersen, Rebecca R. Bellone, Ted Kalbfleisch, Carrie J. Finno.

**Investigation:** Sichong Peng, Erin N. Hales, Jessica L. Petersen, Rebecca R. Bellone, Carrie J. Finno.

**Methodology:** Sichong Peng, Anna R. Dahlgren, Callum G. Donnelly, Erin N. Hales, Jessica L. Petersen, Rebecca R. Bellone, Ted Kalbfleisch, Carrie J. Finno.

**Project administration:** Jessica L. Petersen, Rebecca R. Bellone, Ted Kalbfleisch, Carrie J. Finno.

**Resources:** Jessica L. Petersen, Rebecca R. Bellone, Ted Kalbfleisch, Carrie J. Finno.

**Software:** Ted Kalbfleisch.

**Supervision:** Jessica L. Petersen, Rebecca R. Bellone, Ted Kalbfleisch, Carrie J. Finno.

**Validation:** Jessica L. Petersen, Rebecca R. Bellone, Ted Kalbfleisch, Carrie J. Finno.

**Visualization:** Sichong Peng, Jessica L. Petersen, Rebecca R. Bellone, Carrie J. Finno.

**Writing – original draft:** Sichong Peng.

**Writing – review & editing:** Sichong Peng, Anna R. Dahlgren, Callum G. Donnelly, Erin N. Hales, Jessica L. Petersen, Rebecca R. Bellone, Ted Kalbfleisch, Carrie J. Finno.

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
