## [Decision Letter · Decision Letter 0]

21 Nov 2022

Dear Dr Finno,

Thank you very much for submitting your Research Article entitled 'Functional annotation of the animal genomes: an integrated annotation resource for the horse' to PLOS Genetics.

The manuscript was fully evaluated at the editorial level and by independent peer reviewers. The reviewers appreciated the attention to an important topic but identified some concerns that we ask you address in a revised manuscript.

We therefore ask you to modify the manuscript according to the review recommendations. Your revisions should address the specific points made by each reviewer.

Yours sincerely,

Tosso Leeb

Academic Editor

PLOS Genetics

Gregory Barsh

Editor-in-Chief

PLOS Genetics

Reviewer's Responses to Questions

**Comments to the Authors:**

Reviewer #1: The manuscript presents the first comprehensive Iso-seq transcriptome of the horse. The authors centered around high-quality full-length reads from RNA Iso-seq and used abundant short-read RNA-seq data to refine the annotation of splice junctions and transcription start and termination sites. As a result, the equine transcriptome was improved by adding 39,625 novel transcripts, identifying 84,613 candidate cis regulatory element-gene pairs, and determining gene-isoform ratio of 4.2. As such, the manuscript delivers an awaited and integral resource for the studies of various aspects of equine biology and health by many research groups worldwide.

Comments:

The experimental data presented is of high-quality, the manuscript is well-written, concise and effectively integrates an extensive amount of functional annotation data across the horse genome. I do not have any critical comments and I am looking forward to see this manuscript published. There are just a few minor technical issues to mention:

• Page 14, line 247: remove ‘and’ after (state 4)

• Figure 1: for the ease of understanding, please explain in the legends the acronyms FSM and ISM (even though they are explained in the text)

• Figure 4 right: the figure has ‘uniquitous’, while the legend says ‘ubiquitous’. Please correct.

Reviewer #2: This paper presents a nice contribution to the FAANG international action regarding the annotation of the horse genome. RNA-seq and ATAC-seq experiments were performed on samples from 9 target tissues to identify transcriptomes and chromatin accessibility regions. Previously generated ChIP-seq data was integrated to complement the analysis. The resulting annotation includes gene models, open chromatin regions, Cis Regulatory Elements and epigenetic states of the chromatin. Results are provided as a custom track in the UCSC genome browser for visualization and download.

The article is correctly written and organized. Analyses and results make sense, globally. Considering the available information I do not see any serious issue with the study, but the Methods section should be expanded to clarify a few points of the data analysis.

More specifically, my concerns are the following:

1) The abstract lacks results. It is written like the beginning of an Introduction, or like a longer abstract that has been trimmed. It would be better if it was more condensed and synthetic, in order to lave place to mention the key outcomes and results from the study. This would give more visibility to the article.

2) Methods are not sufficiently detailed. For reproducibility and clarity purpose, the data analysis part of the methods section needs more information, including in particular the version of the tools and the parameters that were used.

A few examples in the "Transcriptome assembly" part:

- line 476: "Clustered transcripts were aligned to the reference genome EquCab3 (2) using minimap2 (63)". Version, parameters?

- l478: "Filtered transcripts from each sample were then merged into a single transcriptome using Cupcake (https://github.com/Magdoll/cDNA_Cupcake/)". Version? Which function/script?

- l480: "and further filtered to retain only those detected in more than one sample." How? Using a Cupcake script or in-house scripts?

etc.

A simple list of tools like in l.492-493 "Data processing, visualization, and statistical analyses were performed using pandas (65), matplotlib (66), seaborn (67), scipy (68), and scikit-learn" is not acceptable. Such phrasing might be used for generic tasks like data visualization or text editing, but data processing and statistical analyses need to be precisely documented. The source code could even be provided, this is common practice.

A useful thought experiment is to imagine the same for the experimental part, for instance if instead of Supp Mat 1 & 2 a single sentence were summarizing "Sample treatments, molecular processing and sequencing experiments were performed using ThermoFisher, Qiagen and Illumina equipments". This would not be considered correct. The same applies to data analysis.

3) In particular, with the exception of the ATAC-seq pipeline, it is not always clear how replicates were handled as they are barely mentioned in the analysis part.

For instance, were the Iso-Seq reads processed for each replicate separately, per tissue or altogether? The experimental description mentions 2 replicates per tissue ("One sample per sex per tissue was selected for sequencing"). Was this information considered? It seems that 4 gtf files are available per tissue on the UCSC browser. If it is the case, how were these generated?

4) Another point that deserves to be clarified and detailed is the way short-read RNA-seq data was used to build the Iso-Seq annotation.

Throughout most of the text, it is presented as if only Iso-seq long read data was used to generate the FAANG annotation, while Illumina short reads were only used for expression quantification. A different strategy is mentioned in the Discussion though. Also, the Methods section describes: "Finally, the total transcriptome was filtered to remove nonsense-mediated decay transcripts, transcripts without short-read coverage support, and transcripts with a splice junction not covered by short-read RNA-seq data".

This looks like a drastic filtering. If true, it means that none of the introns from the final annotation is not supported by short-read data. This raises the question of the specific contribution of the Iso-seq to the assembly in terms of isoform discovery.

I believe that the authors should describe more precisely what they have done and what the outcome was. In particular, how many transcripts were filtered at each step of the analysis.

The choice of ignoring the available short-read RNA-seq data to build the transcriptome using exclusively the Iso-seq data might seem a bit surprising and/or unusual. The difference between the proper pairs ratios in Sup Fig1 vs. Fig2 suggests that a fair amount of valuable information from short read data is missing from the Iso-Seq annotation. Nevertheless, this choice is not a serious issue to me as long as it is clearly presented and documented.

Still, using short read RNA-seq data to validate the Iso-seq assembly like shown in Fig 1D is more problematic to me if this RNA-seq data has already been used to filter the annotation. Not very surprising to get more properly paired reads on Iso-seq given that isoseq transcripts with low coverage have been previously discarded, which was not done for refseq or ensemble annotations. To complement this analysis it could be informative to also provide the proportion of each annotation (ensembl, refseq and isoseq) that is not covered by RNA-seq reads.

5) Differential analyses of gene expression or chromatin accessibility shown in Fig 5 are not described. Required details include software tools, versions, parameters, number of tested genes/peaks, # of replicates, normalization method, model, statistical test, p-value correction, etc.

6) Likewise, chromatin loop prediction should be described in the Methods, better than just "as described by [ref]" l. 284.

Line 289: how are loops "merged"? Given that loops seem to be considered as intervals, is a union performed, an intersection? Would this merging lead to much wider loops, does it ensure that resulting loops still have CTCF peaks at their extremities? L.295, how are loop quantified exactly? Are the read counts from both CTCF peaks added or only the distant one? Which correlation is computed (Pearson, Spearman)? Are raw read counts considered from each sample?

Before considering the correlation between the CRE and the gene expression, wouldn't it make sense to check the chromatin correlation between the promoter vs. CRE peaks of each loop?

Supp Fig 5 is interesting. It is surprising to see such a strong downstream trend. Difficult to estimate if there is a normalization bias. What is the resulting figure when considering all the predicted promoter loops before the correlation filter?

7) Regarding transcript-to-gene ratio in horse vs. human, the authors claim l. 79 that "This difference is largely due to the limitation of the short-read sequencing technology". This is not true. Human or mouse genome annotations have always had more isoforms per gene compared to non-model species, even before long read RNA-seq, and even before RNA-seq technologies altogether. The main reason for a deeper annotation of the human transcriptome is the difference in transcriptomic data quantity. What is currently called "short-read" RNA-seq is not to blame for discovering less isoforms. By the way, the two references that are cited at the end of the sentence do not support that statement. Similarly, the statement "short-read sequencing-based RNA-seq data meant that alternate alternate isoforms could not be accurately resolved" (abstract, l35) is too extreme and should be softened because the current phrasing implies that accurate identification of an isoform is impossible with RNA-seq, which is misleading. Full length isoforms have been correctly identified using "short-read" RNA-seq and confirmed by Sanger sequencing for instance.

8) l. 292: "This was comparable to...". Chromatin loops and TADs are distinct 3D structures, with different scales. This comparison is not really relevant.

9) Is the global FAANG annotation file available somewhere? Maybe this gtf file could be provided as Supplementary Material for an easy access.

10) At the beginning of the CRE section l.228 it would be better to cite the Kingsley et al reference again to remind the readership that the histone data was generated in a previous study, otherwise it is a bit confusing.

Discussion

- The authors mention monoexonic transcripts: what is the proportion of those in the Iso-seq assembly?

- l 332: "our approach used abundant RNA-seq data to refine splice junction, TSS, and TTS annotation." This is not what I saw in the Methods.

- l 343: "While an aggressive approach was taken to ensure 5’ completeness". Which one?

- Just to comment about the DAR/DEG discrepancy in Q2 & Q4, it is true that the dynamic range difference between RNA and DNA quantities is an important factor to consider. Other possible explanations include the time difference between the regulatory action on the DNA and the subsequent effect on the transcriptome, or the simple fact that CREs do not only include enhancers but inhibitors too.

- If the article needs to be shortened, the last paragraph of the Discussion could easily be removed as there is no surprise to find "substantial cell-type differences" between tissues, in particular brain and heart, or testes vs. the others.

Figures

- Shouldn't the units be between parentheses? For instance "Distance to Ensembl TTS (bp)" instead of "Distance to Ensembl TTS, bp".

- Fig 1A: Since the Iso-seq read mapping/processing does not seem to use a reference annotation, how are Iso-seq transcripts assigned to Ensembl TTS? How is it possible to know that the ones several Kb downstream are from the same gene?

- Fig 1B: x-axis label would be more clear with "downstream vs. upstream" or "downstream/upstream ratio" instead of "up- and downstream".

- Was the Fig 1D analysis performed by mapping all the short-read RNA-seq of the entire project each time, including the 57 tissues x 4 replicates, or just a subset?

- Fig. 3B: the figure does not support the statement from the text ("Area under curve (AUC) values of at least 0.6 were achieved for all tissues evaluated (Figure 3B).") because only liver is displayed. Maybe it is possible to show the nine selected tissues, a distribution of the AUC values, or to change the sentence.

Typos:

l.179: "Accuracies of peak calling were using published" [estimated]?

l. 332: "long isoform sequencing" long read?

Reviewer #3: This is a well-written manuscript that reports a comprehensive annotation of the gene regulatory landscape in the horse genome. The annotation resources will be especially valuable to the equine research community, and will also be of broader interest in the area of mammalian gene regulation. The paper is original and does not duplicate previous work. Literature is appropriately cited. The methods are sound and sufficiently described, with some detailed protocols provided in the supplemental material. Original data is deposited in appropriate repositories, and the study conforms to relevant guidelines. The study is very thorough, and I do not have any suggestions for additional analyses.

Minor Comments:

The data and analyses fully support the claims, however not all the numerical data that underlies figures are provided in spreadsheets as supporting information. The only spreadsheets provided are for the GO enrichment and motif enrichment.

Figure 1A and 1B: It would be helpful to state in the Figure legend what FSM and ISM are (even though it’s in the text).

Figure 4: There is a misspelling “Uniquitous”

Supplementary Figure 2 Legend: It seems backwards. Since most of the changes are positive, should it be “Changes in percentages of properly paired reads aligned to Ensembl or RefSeq transcriptomes compared to Iso-seq transcriptome”.

Figures S3 and S4: The explanation of the part with the dots is not clear.

The resources generated would be even more valuable to the research community if the new predicted gene set, or at least the Iso-seq predicted gene set, would be available in gff3 or gtf format, and fasta formatted sequences for transcripts and coding sequence would be made available. I realize the genes are available in BigBed format via UCSC, but converting it to gff3 or gtf might not be straightforward for everyone. Could these files be provided in a data archive site such as Figshare?

**Have all data underlying the figures and results presented in the manuscript been provided?**

Reviewer #1: Yes

Reviewer #2: Yes

Reviewer #3: **No: **Not all the numerical data that underlies figures are provided in spreadsheets as supporting information, but it might not be feasible to provide all of it in spreadsheets.

PLOS authors have the option to publish the peer review history of their article (what does this mean?). If published, this will include your full peer review and any attached files.

Reviewer #1: No

Reviewer #2: No

Reviewer #3: No

---

## [Decision Letter · Decision Letter 1]

28 Jan 2023

Dear Dr Finno,

We are pleased to inform you that your manuscript entitled "Functional annotation of the animal genomes: an integrated annotation resource for the horse" has been editorially accepted for publication in PLOS Genetics. Congratulations!

Yours sincerely,

Tosso Leeb

Academic Editor

PLOS Genetics

Gregory Barsh

Editor-in-Chief

PLOS Genetics

Comments from the reviewers (if applicable):

Reviewer's Responses to Questions

**Comments to the Authors:**

Reviewer #2: The authors addressed most of the points I mentioned (see below). I believe that the abstract and the methods sections in particular needed these improvements. As also suggested by another reviewer, the resulting FAANG annotation is now provided in the form of a downloadable gff3 file, which is likely to be of great value to the community.

- Thanks to the authors for clarifying their Iso-seq filtering strategy using RNA-seq short reads. This filtering method has a strong impact on the results because it implies that no new intron was identified by Iso-seq alone, and that all the splice sites were also found by RNA-seq short reads. Since there is still no gold standard for optimal integration of Iso-seq and RNA-seq data, it is very important that current approaches are clearly document the literature in order to improve future analysis methods. This is another valuable contribution of this paper.

- My concern about using only the % of properly paired RNA-seq reads as a quality metrics (Fig 1D) still stands, because in an extreme case even a randomly generated annotation will tend to 100% if large enough (see the somewhat similar "infinite monkey theorem"). This is why I asked about the proportion of uncovered annotation, to complement sensitivity (or recall) with specificity (or precision). It would have been interesting to have this proportion for the three referred annotations.

- Regarding the CRE loops, I was of course not suggesting to correlate promoters with themselves, but to consider the promoter-CRE chromatin accessibility correlations for the loops that involve a promoter and a regulator. This might have been a relevant positive control.

- Fig. S5 -now S6- still shows a strong unexplained signal: CRE-gene loops are obviously enriched towards the 3' side of the TSS. I have no idea about the reasons for such a bias, if it is a biological feature or an analysis artifact. One hypothesis is that if gene bodies are more GC-rich than intergenic regions they are more likely to contain CTCF binding sites. Comparing the current distribution with the one before filtering by the correlation would have helped to understand if the correlation metrics was partly involved in this bias. It is a bit disappointing that the authors did not comment on that feature.

Other points were correctly addressed.

Congratulations to the authors for that work.

Reviewer #3: The reviewers' comments have been sufficiently addressed in this revision.

**Have all data underlying the figures and results presented in the manuscript been provided?**

Reviewer #2: Yes

Reviewer #3: Yes

PLOS authors have the option to publish the peer review history of their article (what does this mean?). If published, this will include your full peer review and any attached files.

Reviewer #2: No

Reviewer #3: No

**Data Deposition**

http://datadryad.org/submit?journalID=pgenetics&manu=PGENETICS-D-22-01159R1

**Press Queries**

---

## [Editor Report · Acceptance letter]

27 Feb 2023

PGENETICS-D-22-01159R1 

Functional annotation of the animal genomes: an integrated annotation resource for the horse 

Dear Dr Finno, 

We are pleased to inform you that your manuscript entitled "Functional annotation of the animal genomes: an integrated annotation resource for the horse" has been formally accepted for publication in PLOS Genetics! Your manuscript is now with our production department and you will be notified of the publication date in due course.

With kind regards,

Anita Estes

PLOS Genetics

On behalf of:
